# Amorphization of Ethenzamide and Ethenzamide Cocrystals—A Case Study of Single and Binary Systems Forming Low-Melting Eutectic Phases Loaded on/in Silica Gel

**DOI:** 10.3390/pharmaceutics15041234

**Published:** 2023-04-13

**Authors:** Katarzyna Trzeciak, Ewelina Wielgus, Sławomir Kaźmierski, Tomasz Pawlak, Marek J. Potrzebowski

**Affiliations:** Centre of Molecular and Macromolecular Studies, Polish Academy of Sciences, Sienkiewicza 112, 90-363 Lodz, Poland; katarzyna.trzeciak@cbmm.lodz.pl (K.T.); ewelina.wielgus@cbmm.lodz.pl (E.W.); slawomir.kazmierski@cbmm.lodz.pl (S.K.); tomasz.pawlak@cbmm.lodz.pl (T.P.)

**Keywords:** amorphization, melting, ball milling, physical mixture, dissolution kinetics, solid state NMR, powder X-ray diffraction (PXRD), differential scanning calorimetry (DSC)

## Abstract

The applicability of different solvent-free approaches leading to the amorphization of active pharmaceutical ingredients (APIs) was tested. Ethenzamide (ET), an analgesic and anti-inflammatory drug, and two ethenzamide cocrystals with glutaric acid (GLU) and ethyl malonic acid (EMA) as coformers were used as pharmaceutical models. Calcinated and thermally untreated silica gel was applied as an amorphous reagent. Three methods were used to prepare the samples: manual physical mixing, melting, and grinding in a ball mill. The ET:GLU and ET:EMA cocrystals forming low-melting eutectic phases were selected as the best candidates for testing amorphization by thermal treatment. The progress and degree of amorphousness were determined using instrumental techniques: solid-state NMR spectroscopy, powder X-ray diffraction, and differential scanning calorimetry. In each case, the API amorphization was complete and the process was irreversible. A comparative analysis of the dissolution profiles showed that the dissolution kinetics for each sample are significantly different. The nature and mechanism of this distinction are discussed.

## 1. Introduction

Knowledge of the dissolution profile of active pharmaceutical ingredients (APIs) under physiological conditions is one of the key factors determining the possibility of using drugs in medical practice. It is well known that over 70% of drugs and new drug candidates are poorly soluble in water, causing their low bioavailability and a challenge in the development of new dosage forms [1]. In oral treatment, the control of API release kinetics from tablets is one of the most important parameters determining the therapeutic strategy. A rapid release is not always the desired effect, especially when the concentration of API in the blood should be maintained for a long time. Thus, various approaches have been widely tested by scientists involved in the works on the formulation of APIs [2,3,4,5,6,7]. These works include the formation of cocrystals [8,9], salt formation [10,11,12], and the use of drug delivery systems [13,14,15,16,17,18].

Very recently, the formation of pharmaceutical cocrystals has attracted much attention as a method for controlling the physicochemical properties of drugs [19]. According to the generally accepted definition, cocrystals are homogenous (single-phase) crystalline structures that are made up of two or more components in a definite stoichiometric ratio, where the arrangement in the crystal lattice is not based on ionic bonds (as with salts) [20,21]. The remarkable advantage of pharmaceutical cocrystals is their significant improvement in terms of physicochemical properties without compromising on therapeutic benefits. Usually, therapeutic cocrystals consist of two components, an API and a complementary molecular coformer, although API:API cocrystals are also known [22,23]. For example, this group includes Entresto^®^, a cocrystal of the drugs valsartan and sacubitril used to treat heart failure [24]. The basic requirement for a suitable coformer is to be pharmaceutically acceptable, i.e., generally regarded as safe (GRAS) substances [25]. Coformers should be relatively cheap, with a rather low molecular weight, and possess multiple API-binding sites that can be involved in the formation of strong intermolecular interactions. The vast majority of known cocrystals are characterized by better solubility in water and gastric fluid compared to pure APIs [26,27].

It is well known that the solubility of any solid sample strongly depends not only on its chemical composition but also on its physical state. Differentiated solubility can be expected when the sample is crystalline or amorphous. It has been proven that amorphous forms of drug systems have a higher solubility compared to the pure crystalline form. Today, different amorphization methods are used in pharmaceutical practice [28]. One of them is the amorphous solid dispersion (ASD) technology, which utilizes inert carriers (mostly polymers) to disperse the amorphous drugs and prevent their recrystallization [29,30,31]. An alternative approach to the conventional polymer amorphous solid dispersion (PASD) [32] is the formation of co-amorphous systems [33,34,35]. The term ‘co-amorphous’ was introduced by Chieng et al. and defines the amorphous systems composed of substances with a low molecular weight [36]. Another commonly used method is based on the use of mesoporous particles, which have recently gained considerable attention as a stabilizer for amorphous formulations [37,38,39]. The majority of the mesoporous particles utilized in the pharmaceutical field are mesoporous silica nanoparticles such as MCM-41 [40,41,42,43,44], Neusilin^®^ [45,46], UFL2 [47], Fujicalin^®^ [48,49], FujiSil^TM^ [50] and SBA-15 [51,52,53].

The aim of our work was to test several approaches that can be used to control the solubility of APIs. These approaches include the formation of cocrystals, the preparation of co-amorphous solids by solvent-free methods (manual mixing, thermal method, and ball milling), and the use of silica particles as a medium to support amorphization. The problem of the spontaneous crystalline-to-amorphous phase transformation of medicinal compounds in the presence of silica porous media was exhaustively discussed by Bogner and co-workers [54,55,56]. In cited papers, the amorphization capacity, mechanisms of interactions, and thermodynamics of these processes were investigated by means of nitrogen adsorption–desorption and powder X-ray diffraction techniques. The authors concluded that the amorphous capacity is correlated with surface curvature and is facilitated by capillary condensation and further enhanced by dipole–dipole or dipole-induced interactions, promoted by hydroxyl groups on the SiO_2_ surface.

As model samples for studies, we employed ethenzamide (ET) [57] and its cocrystals with ethylmalonic acid (EMA) and glutaric acid (GLU) as coformers. Ethenzamide (ET), a common analgesic and anti-inflammatory drug, is known to form low-melting cocrystals with glutaric acid (GLU) (TM = 76 °C) [58] and two cocrystal polymorphs with ethylmalonic acid (EMA) (TM of ca. 85 °C) [59]. The neat grinding and hot melting of ET with either GLU or EMA leads to ET:GLU and ET:EMA cocrystals, respectively, as shown before in the literature. In both methods, the obtained cocrystals have a similar structure, as was confirmed by sc-XRD, PXRD, and solid-state NMR measurements. The single crystals’ X-ray data for ET and the ET:EMA and ET:GLU cocrystals were deposited in CSD (VAKTOS, VAKTOS01, TIWPIB).

## 2. Materials and Methods

### 2.1. Materials

Ethenzamide (ET), ethylmalonic acid (EMA), and glutaric acid (GLU) were obtained from Sigma-Aldrich and used without further purification. Silica gel (SiO_2_) 60 (0.040–0.063 mm, 230–400 mesh ASTM), with a pore volume of 0.74–0.84 mL/g and surface area of 480–540 m^2^/g, was purchased from Merck and was activated by calcination at 300 °C for 1 h to remove the water (SiO_2(calc)_) or used without thermal treatment (SiO_2(hydr)._)

### 2.2. Cocrystals Preparation

The ET:GLU and ET:EMA cocrystals were prepared using the mixer mill MM200. The physical mixtures of ET:GLU and ET:EMA in 1:1 molar ratios were ground separately in the steel jars (10 mL) with one ball (Ø = 10 mm) for 1 h with an oscillation frequency of 25 Hz. All substances were micronized in a mortar before grinding.

### 2.3. Manual Mixing (MM)

The mixtures of ET or cocrystals and SiO_2(calc)_ or SiO_2(hydr)_ (1:3 *w*/*w*) were gently stirred by hand with a spatula, and then the mixture was inserted into an Eppendorf vessel and shaken for 1 min. The ratio of API to silica (*w*/*w*) is an arbitrary choice.

### 2.4. Thermal Method (TM)

The mixtures of ET or cocrystals and SiO_2(calc)_ or SiO_2(hydr)_ (1:3 *w*/*w*) were heated above their melting point for 50 min. The heating temperatures for cocrystals and ET were 90 °C and 135 °C, respectively. The ratio of API to silica (*w*/*w*) is an arbitrary choice.

### 2.5. Ball Milling Method (BM)

The mixtures of ET or cocrystals with SiO_2(calc)_ or SiO_2(hydr)_ (1:3 *w*/*w*) were ground separately in steel vessels (12 mL) with ten balls (Ø = 5 mm) for 70 min at 450 rpm using a planetary ball mill PM 200. The ratio of API to silica (*w*/*w*) is an arbitrary choice.

### 2.6. PXRD Measurements

Panalytical Empyrean powder diffractometer was used in the collection of diffraction data on the powder samples. The samples were analyzed in Bragg–Brentano reflection mode using Cu-Kα radiation (λ = 1.5419 Å), with the 2Θ range of 5–35° in three continuous scans using 0.0131° step size. For the incident beam, a 0.02 rad. Soller slit, fixed divergence slit of 1/8° and a fixed mask of 10 mm were used. For data collected under non-ambient conditions, the instrument was equipped with CHC plus Anton Paar chamber. Obtained scans were tested for any discrepancies suggesting adverse reaction of the sample to X-ray irradiation and summed up.

### 2.7. Solid-State NMR Experiments

All solid-state NMR experiments were registered on a Bruker Avance III 600 spectrometer, with an operational frequency of 150.93 and 600.15 MHz for ^13^C and ^1^H, respectively. In each experiment, a sample was placed in a 4 mm ZrO_2_ rotor and spun with an 8 or 12 kHz spinning speed for ^1^H MAS [60] and ^13^C CP/MAS and ^13^C SPE/MAS, respectively, with a repetition time of 10 and with the measurement temperature kept at 25 °C, unless specified otherwise. A sample of U-^13^C, ^15^N-labeled histidine hydrochloride was used to set the Hartmann–Hahn condition for ^13^C, with a proton 90° pulse length of 4 μs. For cross-polarization, the nutation frequency was 50.5 kHz for ^13^C, with a ^1^H ramp shape from 90% to 100% and a ^1^H nutation frequency of 62.5 kHz. In all cases, SPINAL-64 decoupling sequence [61] with a ^1^H pulse length of 3.6 μs was applied. The ^13^C chemical shift was referenced indirectly by using adamantane (resonances at 38.48 and 29.46 ppm) as an external secondary reference [62]. The Pb(NO_3_)_2_ was used for temperature calibration.

### 2.8. Differential Scanning Calorimetry

Calorimetric measurements were performed using a TA Instruments Trios V5.1.0.46403 apparatus in nitrogen flow. Samples with ET:GLU and ET:EMA cocrystals and ET were heated from room temperature to 100 °C or 150 °C, respectively. The samples were then cooled to 0 °C and heated to 100 °C or 150 °C at a heating and cooling rate of 10 °C/min.

### 2.9. Dissolution

In vitro dissolution studies were carried out using USP II paddle method on dissolution tester (model—Vision G2 Classic 6, Hanson). In all samples, the amount of ET was 15 mg. The samples were placed in 900 mL of Milli-Q water (pH 5.7) and simulated gastric fluid without pepsin (SGFsp, pH 1.2). SGFsp was prepared by adding 2 g NaCl and 7 mL concentrated HCl to 1000 mL distilled water. The temperature of the dissolution medium was maintained at 36 °C ± 0.5 and the rotation speed was kept at 25 rpm. Aliquot samples of 1 mL were withdrawn at predetermined time intervals (1, 2, 4, 6, 8, 10, 15, 20, 30, 40, 50, 60, 75, 95, and 105 min) and equivalent volumes of fresh water were added to keep a constant dissolution volume. Samples were filtered using a 0.45 μm PTFE syringe filter.

The concentration of dissolved ET was determined by ACQUITY UPLC I-Class chromatography system equipped with a photodiode array detector (Waters Corp., Milford, MA, USA). Chromatographic separation was attained on an ACQUITY UPLC™ BEH C18 column (100 × 2.1 mm, 1.7 μm). The optimal absorption wavelength for ET was determined and set at 235 nm. The flow rate was 0.35 mL/min, the column temperature was 60 °C and the injection volume was 1 μL.

A gradient program was employed with the mobile phase combining solvent A (0.1% formic acid) and solvent B (methanol) as follows: 35% B (0–0.5 min), 35–85% B (0.5–2.8 min), 85–85% B (2.8–3.8 min), 85–35% B (3.8–3.9 min), and 35–35% B (3.9–5.5 min). The initial stock calibration solution of ET was created with a concentration of approximately 10 mg/mL in methanol and stored at 4 °C. The stock solution was serially diluted (with methanol/water (50:50, *v*/*v*)) to obtain working solutions at several concentration levels.

Two calibration curves were prepared at six different concentrations of ET solutions in the range from 0.7 to 25 mg/L, with correlation coefficients above 0.999. The system was controlled by using MassLynx software (Version 4.1), and data processing (peak area integration, construction of the calibration curve) was performed by the TargetLynxTM program.

## 3. Results

Silica gels are often used as an adsorbent as a stationary phase in chromatography and as a carrier in catalytic processes. According to the US Food and Drug Administration and the European Food Safety Authority, amorphous forms of silica and silicates are also approved as active media in the pharmaceutical industry [63]. This material is generally recognized to be safe as oral delivery ingredients in amounts of up to 1500 mg per day [64]. Silica is used in the formulation of solid dosage forms, e.g., tablets, glidants, lubricants, or amorphizing agents.

As we highlighted in the Introduction due to the presence of hydroxyl groups, the oxygen atoms visible on the surface of silica gel particles are coupled with protons, making the surface of silica gel extremely polar. Silica gels are considered to be highly hydrophilic and spontaneously absorb water, developing a thick hydration layer. The role of surface waters (Figure 1) in the spontaneous process of amorphization was recognized as a relevant factor by Bogner et al. [56].

### 3.1. Solid-State NMR, Differential Scanning Calorimetry (DSC), and Powder X-ray Diffraction (PXRD) Studies of Ethenzamide with Silica

#### 3.1.1. Analysis of Physical Mixture

We began our studies by determining the degree of hydration of silica used in further studies. For this purpose, we used thermogravimetric analysis (TGA). The TGA profile (attached in the Appendix A) showed that the SiO_2_ sample contained approximately 7% of the water on the silica surface. The water content of silica is not constant and may vary depending on the humidity environment. According to the manufacturer’s information, this content can reach a value of up to 9%.

The ^1^H MAS NMR measurements were employed for the qualitative inspection of silica samples. Figure 2a shows the spectrum recorded with a spinning rate of 8 kHz for calcinated silica kept 1h in the oven at 300 °C. Figure 2b displays the spectrum recorded under the same conditions for sample stored in a natural environment without special humidity control. The difference between both spectra is apparent. The ^1^H chemical shift of Si-OH protons for calcinated silica is equal to δ_1H_ = 2.1 ppm, whereas, for the sample containing water molecules on the surface, the δ_1H_ is 4.0 ppm. Unexpected differences in spectra were noted for physical mixtures of ethenzamide (ET) with calcinated silica and hydrated silica. Figure 2c shows the ^1^H MAS spectrum recorded with a spinning rate of 8 kHz for mixture of ET/SiO_2(calc)_ with a weight-to-weight ratio of 1:3. This spectrum presents a typical solid-state pattern with very broad lines, the visible overlapped two components representing Si-OH and ET protons. In contrast, the spectrum shown in Figure 2d representing the mixture of ET/SiO_2(hydr)_ is significantly different and has features typical for “liquid-like solids”, with clearly seen relatively sharp resonance lines of ET. Such a strong effect can be explained by assuming the hydrophilic interactions between ET and the surface water. It is very likely that ET molecules, during contact with SiO_2(hydr)_, are located in “water baths” and behave as molecules in a dense solution with restricted mobility. The further proof confirming the mobility of ET on the SiO_2_ surface is the ^13^C CP/MAS experiment. It is commonly accepted that the CP experiment, which is based on ^1^H-to-^13^C magnetization transfer driven by ^1^H−^13^C heteronuclear dipolar interactions, is not able to detect ET in contact with water molecules because of the high mobility of ET. Fast molecular motion averages out this key interaction, and therefore the CP technique is only sensitive to rigid molecules. The single-pulse experiment (SPE) MAS is an alternative option that allows for the measurement of mobile phases. Figure 2e shows the ^13^C CP/MAS spectrum, while Figure 2f displays the ^13^C SPE/MAS spectrum for the ET/SiO_2(hydr)_ sample. The difference between the spectra is obvious, and this applies to both the spectral pattern, chemical shifts, and the width of the resonance lines. This confirms that ET mixed with SiO_2_ is located in different zones and has distinct molecular dynamics.

We further observed the difference between the ET/SiO_2(calc)_ and ET/SiO_2(hydr)_ systems by analyzing their thermal properties in the temperature range from 22 °C to 90 °C. It is worth reminding that the melting point of pure ET is equal to 132–134 °C. The variable temperature (VT) ^1^H MAS spectra of ET crystalline solid are shown below (Figure 3).

As seen, the ^1^H MAS NMR spectra of ET with an increase in temperature up to 90 °C are very similar, which means that melting or phase transition processes do not occur during heating and sample spinning in the zirconia rotor. The ^1^H MAS spectra for ET mixed with silica are significantly different (see Figure 4). The left column shows spectra for ET/SiO_2(calc)_ whereas the right column displays spectra for ET/SiO_2(hydr)_. In the middle space, the temperature of the measurement is defined. From a comparative analysis of the spectra, it is concluded that water molecules play an important role in increasing the mobility of ET. The resolution of the proton spectrum for ET/SiO_2(calc)_ at 70 °C is comparable with the spectrum of ET/SiO_2(hydr)_ at 40 °C. Moreover, the resonance lines for ET/SiO_2(hydr)_ are sharper compared to ET/SiO_2(calc)_. At 90 °C, the full-width at half-height (FWHH) measured on methyl signals was found to be 130 Hz for ET/SiO_2(calc)_, whereas, for ET/SiO_2(hydr)_, it is equal to 98 Hz.

The solid-state NMR data are consistent with powder X-ray diffraction (PXRD) measurements and DSC studies (Figure 5). Figure 5a shows the PXRD diffractogram of physical mixture ET/SiO_2(calc)_. Figure 5b shows a diffraction of pure ET. The similarity of the diffractograms means that, immediately after mixing the two components, the crystallinity of ET is preserved. However, it should be emphasized that the diffractogram analysis does not allow for a quantitative analysis and determination of the ratio of the amorphous-to-crystalline phase. The DSC profile of pure crystalline ET is shown in Figure 5c. The blue dashed line represents the first run in the temperature range from 0 °C to 150 °C, with the melting temperature (endothermic peak) at 132 °C. The green line displays the profile for the cooling process. The very strong and sharp exothermic peak shows the recrystallization of ET at a temperature equal to 115 °C. The second run (red line) is identical to the first run and both profiles overlap.

The case of ET/SiO_2(calc)_ is different (Figure 5d). The first run reveals that the endothermic transition begins early, at 30 °C. The sharp endothermic peak at 132 °C represents the melting of trace amounts of crystalline ET, suggesting that amorphization is a continuous process and takes time. No ET recrystallization is observed during cooling (green line). The second run (red line) proves that there is no crystalline phase ET in the sample and that the amorphization is complete.

Summarizing the results presented in this section, we can conclude that ET interacts strongly with silica, even without external stimuli (thermal, mechanical). When manually mixing two components, the ET part is amorphized and the part retains its crystal structure. Full ET amorphization can be achieved by a long storage of the sample (spontaneous amorphization), melting, or grinding in a ball mill. However, particular care should be taken when using the melting method, as ET sublimates easily. A similar problem was identified during the melting of naphthalene with SiO_2_ [56].

#### 3.1.2. Analysis of ET/SiO_2_ Systems Melted or Ground in a Ball Mill

The results presented in Figure 5d were further verified by employing the diffraction technique. Figure 6a shows the variable temperature (VT) PXRD measurements of ET/SiO_2(hydr)_. The sample was measured in the temperature range from 20 °C to 120 °C. The diffractogram at 20 °C has already been discussed in the previous section. An analysis of PXRD data shows that the bulge of the baseline in the range 2Θ 18–25, representing the amorphous phase, increases with an increasing temperature. At 80 °C, the reflexes coming from the crystalline ET phase are very weak. At 100 °C, only the bulge baseline is observed. The sample cooled down to room temperature does not show any reflexes representing crystalline ET, which means that amorphization is completed. It is worth noting that pure ET behaves dramatically differently (see Figure 5c). Figure 6b displays the diffractograms at temperatures of 20 °C and 120 °C. Even at a high temperature of 120 °C, no melting or phase transition was observed. In addition, we observed that the molten ET sample easily crystallizes at room temperature, forming crystals with a morphology known from X-ray studies of single crystals. Finally, as we mentioned above, ET is also one of the materials that sublimate, and weight loss is noted when heated. By using silica as an amorphizing agent due to the decrease in the thermal transition temperature, the API is partially protected against this effect.

Another method that we used for ET/SiO_2(hydr)_ amorphization was neat grinding, employing the planetary ball mill PM200. The physical mixture of two components was mixed for 70 min in a ball mill at 450 rpm throughout the process. Figure 7 shows the compiled information about the state of matter for ET/SiO_2_ coming from measurements when different instrumental techniques are applied. From the analysis of the PXRD diffractogram displayed in Figure 7a, it is clear that the ET/SiO_2(hydr)_ sample is amorphous, where only very fine traces of the crystalline phase are visible. This conclusion is consistent with DSC studies shown in Figure 7b. Figure 7c–e present ^1^H MAS, ^13^C CP MAS, and ^13^C SPE MAS spectra, respectively. In particular, the ^13^C NMR results are informative. Figure 7d proves that traces of the crystalline phase remain in the grounded mixture. It is worth noting that the amorphous phase (Figure 7e) is represented by different spectral patterns compare to the crystalline sample. The resonance lines are very broad and located in different positions.

### 3.2. Silica-Particles-Based Amorphization of ET:EMA and ET:GLU Cocrystals

Figure 8 shows the ^13^C solid state NMR spectra for samples in a different stage of preparation. Figure 8a displays the ^13^C CP/MAS spectrum of the physical mixture of ET and GLU. Figure 8b presents the ^13^C CP/MAS spectrum of the well-defined ET:GLU cocrystal. Figure 8c shows the ^13^C CP/MAS spectrum of the physical mixture ET:GLU cocrystal and SiO_2(hydr)_ with a weight-to-weight ratio of 1:3. The ^13^C SPE/MAS spectrum of the ET:GLU/SiO_2(hydr)_ physical mixture (1:3) is shown in Figure 8d. Comparing these four spectra, one can see that each case is different. Figure 8c,d represent the crystalline and amorphous phases of ET:GLU deposited on SiO_2(hydr)_. The most striking difference is seen in the carboxylic region. In the case of the cocrystal, carboxylic residues involved in the formation of a structural quadruplex via hydrogen bonding −N-H^….^O=C-O and O-C=O^……^H-O-C=O are represented by resonances at δ = 182.0 ppm and δ = 177.5 ppm, respectively. In the case of the amorphous phase, these signals are averaged (δ = 180.1 ppm). The difference in the number of NMR signals is also visible in the aliphatic region. In Figure 8c, the resonances of the methyl groups are doubled, which suggests the presence of amorphous and crystalline phases.

Figure 9 shows the PXRD, DSC, and VT NMR data for the physical mixture of the ET:GLU cocrystal and SiO_2(hydr)_. Diffractograms shown in Figure 9a reveal that the freshly mixed sample contains the crystalline ET:GLU components and amorphous phase; however, after melting and cooling, the sample is amorphous, as shown by DSC studies. (Figure 9b). The first run (blue line) shows the melting of the cocrystal at 69 °C. During the cooling (green line), the recrystallization of ET:GLU is not observed. The full amorphization is confirmed by the second run (red line). ^1^H MAS (Figure 9c, left column) and ^13^C SPE MAS (Figure 9c, right column) provide further evidence showing that, when increasing the temperature from 30 °C to 80 °C, the cocrystal is in a molten state.

The amorphization of the sample was also achieved when a mixture of the ET:GLU cocrystal and SiO_2(hydr)_ was neatly grinded using the ball mill. Figure 10a shows the PXRD diffractogram, and Figure 10b shows DSC profiles. The NMR spectra are shown in Figure 10c (^1^H MAS), Figure 10d (^13^C CP/MAS), and Figure 10e (^13^C SPE MAS). All presented data clearly prove that the grinding procedure leads to amorphization.

A similar methodology was used to study the ET:EMA cocrystal mixed with SiO_2(hydr)_. The behavior of ET:EMA in contact with silica is very similar and consistent with that of ET:GLU. The full set of experimental data confirming this conclusion is attached as Appendix A.

### 3.3. Dissolution Studies

The main objective of the project is to answer the question of how API amorphousness loaded on/in silica and forced by different technical approaches affects the kinetics of dissolution. For this purpose, we used several samples, starting with pure components ET, ET:GLU, and ET:EMA. Measurements were made at 36 °C in Milli-Q water (pH 5.7) and in simulated gastric fluid without pepsin (SGFsp, pH 1.2). The amount of ET for each sample was 15 mg. Figure 11a shows kinetic profiles measured over a time range of up to 105 min (pH 5.7). From the analysis of the profiles, it is seen that pure ET dissolves faster than ET from the cocrystals ET:GLU or ET:EMA. After 20 min, 86.2% of ET, 63.5% ET from ET:GLU, and 52.4% of ET from ET:EMA are dissolved. We obtained this interesting information by comparing dissolution curves (Figure 11b) for samples of ET mixed with silica (ET/SiO_2_/MM), melted with silica (ET/SiO_2_/TM), and ground with silica (ET/SiO_2_/BM). As one can see, in the formed physical mixture, the solubility of ET is significantly increased, particularly in the first dissolution period up to 10 min. After 6 min, 83.4% of ET is dissolved from ET/SiO_2_/MM, whereas, in the case of ET/SiO_2_/TM and ET/SiO_2_/BM, these values are 38.6% and 59.4%, respectively. It is worth stressing that, after 6 min, the concentration of ET for ET/SiO_2_/MM is almost two times larger compared to pure ET (43.6%).

Figure 12 shows the kinetic profiles of samples dissolved in simulated gastric fluid without pepsin (SGFsp) with a pH of 1.2. As can be seen, the trend is very similar to that shown in Figure 11, although the distinction between the dissolution rate for ET and ET:GLU, ET:EMA cocrystals is smaller (Figure 12a). As in the previous case, the release of ET from the ET/SiO_2_/MM sample is faster compared to other samples (Figure 12b)

Figure 13a shows dissolution profiles of ET for ET:GLU/SiO_2_/MM, ET:GLU/SiO_2_/TM, and ET:GLU/SiO_2_/BM samples (water, pH 5.7). Each curve representing the ET component of mixtures illustrates different dissolution kinetics. As in the previous case, the rate of dissolution of ET from the ET:GLU cocrystal in the physical mixture with silica (ET:GLU/SiO_2_/MM) is much higher compared to other samples (ET:GLU/SiO_2_/TM or BM). After 6 min, 83.1% of ET is dissolved from ET:GLU/SiO_2_/MM, whereas, in the case of ET:GLU/SiO_2_/TM and ET:GLU/SiO_2_/BM, these values are 32.7% and 51.9%, respectively. It is worth noting that the melting method (sample ET:GLU/SiO_2_/TM) can be considered as a procedure leading to a slowdown in the dissolution kinetics. The analysis of kinetic profiles for ET:EMA/SiO_2_/MM, ET:EMA/SiO_2_/TM, and ET:EMA/SiO_2_/BM samples shown in Figure 13b leads to a similar conclusion as presented *supra*. The ET in the ET:EMA/SiO_2_/MM sample is dissolved very quickly: after 6 min, 76.0% is released from the solid. There is a significant difference in the solubility of the ET from thepure cocrystal ET:EMA and from the cocrystal ET:EMA associated with SiO_2_, regardless of the method of preparation of ET:EMA/SiO_2_ samples. After 6 min, only 15.7% of ET is detected in the solution, whereas, for sample ET:EMA/SiO_2_/TM, this value is 55.4% and, for ET:EMA/SiO_2_/BM, it is 49.4%.

Finally, it should be stressed that the change in environmental conditions does not significantly affect the release profiles. At pH 1.2 (Figure 14), similar trends were observed to when at pH 5.7. The release of ET from the ET:GLU/SiO_2_/MM and ET:EMA/SiO_2_/MM is faster compared to other samples.

## 4. Conclusions

In this work, we tested three solvent-free approaches leading to the amorphization of active pharmaceutical ingredients (APIs). The ethenzamide and two ethenzamide cocrystals with glutaric acid and ethyl malonic acid as coformers were used as pharmaceutical models. We proved that, in any case, silica gel is an effective carrier that facilitates amorphization. Our current studies complement our previous investigations related to applications of mesoporous silica particles as a drug delivery system. In a recent paper using MCM-41, we pointed out that, in addition to coformer selection, the method used for loading is an important factor that can fine-tune the release kinetics of guest particles. It was found that the loading method may have a more decisive impact on the release rate than the composition of binary systems [42].

In the current project, we investigated calcinated and thermally untreated silica gels to see what the influence of surface water is on the spontaneous amorphization of APIs. It was found that surface water can be considered as an assistant of amorphization, which accelerates this process rather than slowing it down as in the case of naphthalene [56]. This observation is important in light of dissolution studies that are carried out in the aquatic environment. It is worth emphasizing that, depending on the methods used for sample amorphization, we observed different dissolution profiles. For samples of ET and ET:GLU and ET:EMA cocrystals manually mixed with SiO_2_, ET dissolves very quickly, and much faster than pure samples. In the case of melting or grinding the ET, dissolution is much slower. This phenomenon can be explained if we assume that manual mixing leads to API interactions with SiO_2_ on the surface, and the strength of these interactions is weakened by surface waters. In such an environment, the API is very mobile and ready for migration. The situation in which melting or grinding methods are used is different. API under the influence of thermal or mechanical stimuli is located in the space of silica gel under the surface, in pores or voids. The interaction of the API with silica in the pores (voids) is stronger compared to the surface interactions of the API, as a result of which dissolution is slower. It is interesting to note that the slowest dissolution we observed was for the ET:EMA cocrystal. A comparative analysis of the dissolution profiles showed that the dissolution kinetics can be controlled not only by the composition of the API (cocrystals) but also by the amorphization method. Such a distinction may be an important feature to consider when planning a therapeutic strategy.

## Figures and Tables

**Figure 1 pharmaceutics-15-01234-f001:**
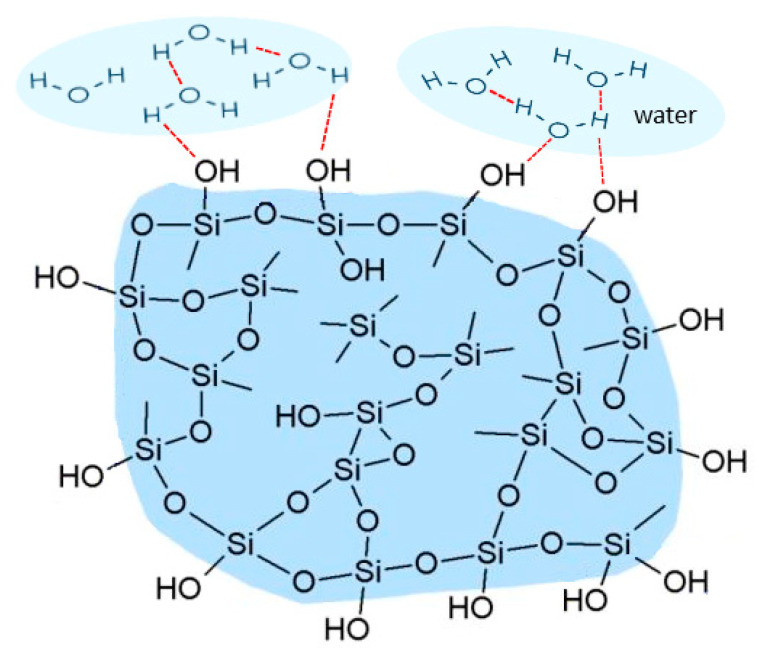
Schematic illustration of the silica gel structure with the hydrogen bonding shown in red.

**Figure 2 pharmaceutics-15-01234-f002:**
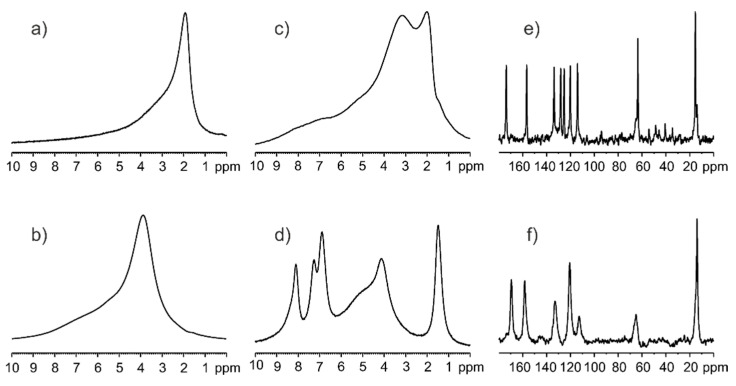
^1^H MAS NMR spectra for (**a**) calcinated silica kept 1h in the oven at 300 °C, (**b**) silica stored in a natural environment without special humidity control, (**c**) mixture of ET/SiO_2(calc)_ with weight-to-weight ratio 1:3, (**d**) mixture of ET/SiO_2(hydr)_ with weight-to-weight ratio 1:3 recorded with a spinning rate of 8 kHz. The ^13^C CP/MAS (**e**) and ^13^C SPE/MAS spectrum (**f**) for ET/SiO_2(hydr_) sample were recorded with a spinning rate of 12 kHz.

**Figure 3 pharmaceutics-15-01234-f003:**
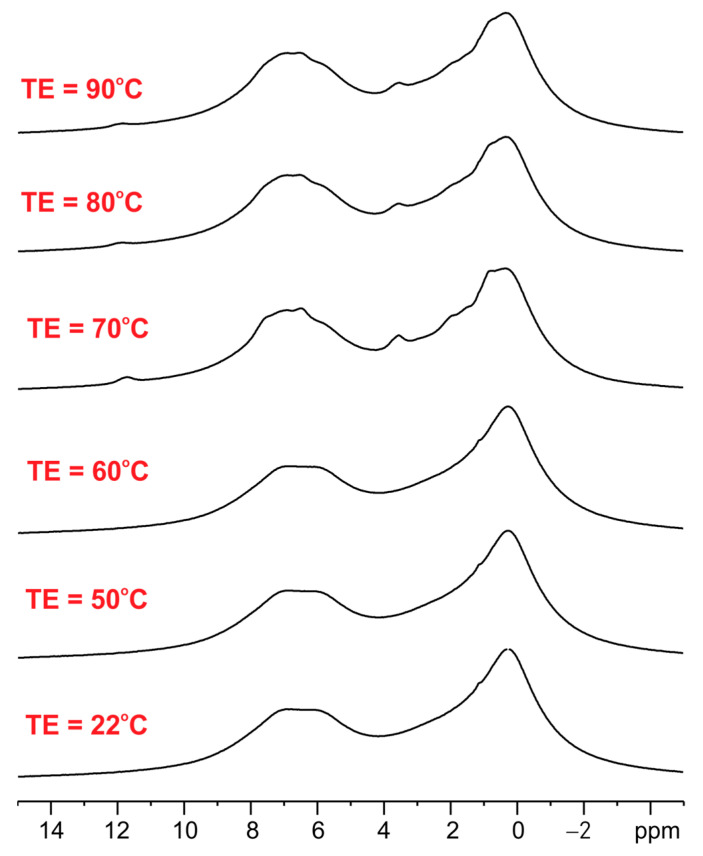
The variable temperature (VT) ^1^H MAS spectra of ET crystalline solid recorded with a spinning rate of 8 kHz.

**Figure 4 pharmaceutics-15-01234-f004:**
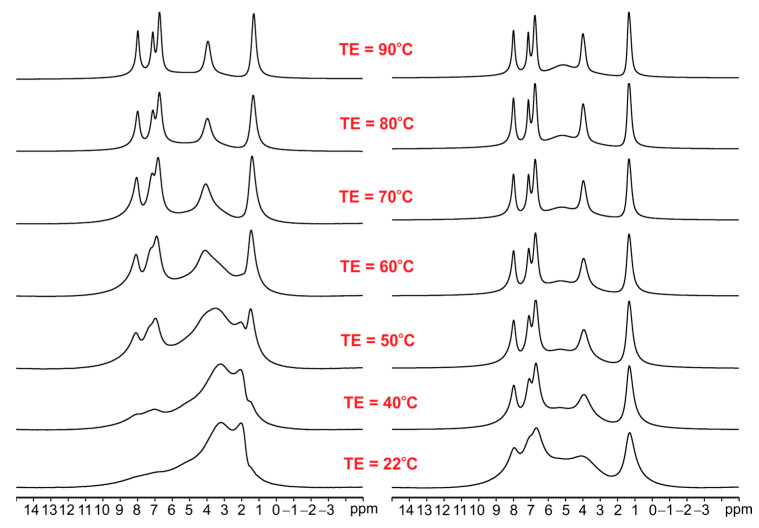
The variable temperature (VT) ^1^H MAS spectra of ET/SiO_2(calc)_ (**left column**) and ET/SiO_2(hydr)_ (**right column**) recorded with a spinning rate of 8 kHz.

**Figure 5 pharmaceutics-15-01234-f005:**
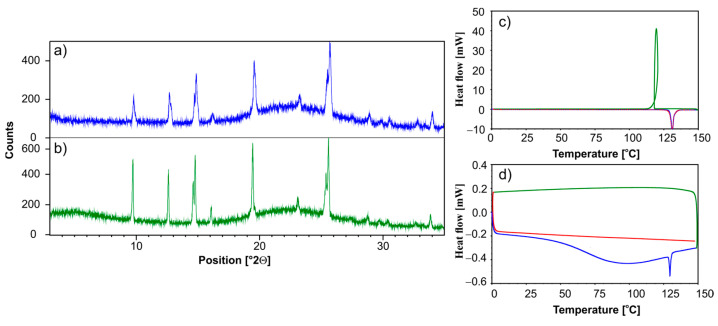
PXRD of (**a**) physical mixture ET/SiO_2(calc)_, (**b**) physical mixture ET/SiO_2(hydr)_ recorded on a PANalytical 3 kW system in Bragg–Brentano geometry and with a Cu Kα (λ = 1.5425 Å) source. The DSC measurements of (**c**) ET and (**d**) physical mixture ET/SiO_2(hydr)_. The first heating run from 0 °C to 150 °C (blue line), then a cooling run from 150 °C to 0 °C (green line), followed by a second heating run from 0 °C to 150 °C (red line).

**Figure 6 pharmaceutics-15-01234-f006:**
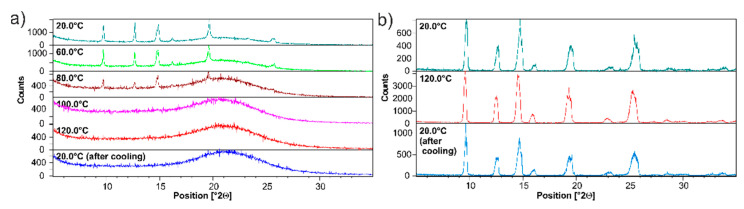
PXRD of (**a**) physical mixture ET/SiO_2(hydr)_, (**b**) pure ET in the range of temperatures (20–120 °C). Recorded on a PANalytical 3 kW system in Bragg–Brentano geometry and with a Cu Kα (λ = 1.5425 Å) source equipped with CHC plus Anton Paar chamber.

**Figure 7 pharmaceutics-15-01234-f007:**
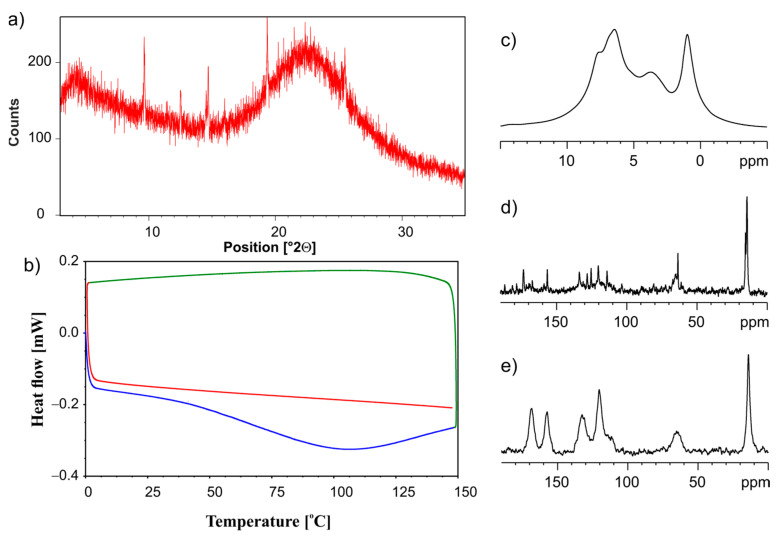
(**a**) PXRD recorded on a PANalytical 3 kW system in Bragg–Brentano geometry and with a Cu Kα (λ = 1.5425 Å) source, (**b**) the DSC measurement. The first heating run from 0 °C to 150 °C (blue line), then a cooling run from 150 °C to 0 °C (green line), followed by a second heating run from 0 °C to 150 °C (red line), (**c**) ^1^H MAS, (**d**) ^13^C CP MAS, and (**e**) ^13^C SPE MAS NMR measurements recorded with a spinning rate of 12 kHz for the physical mixture of ET/SiO_2(hydr)_ ground in a ball mill.

**Figure 8 pharmaceutics-15-01234-f008:**
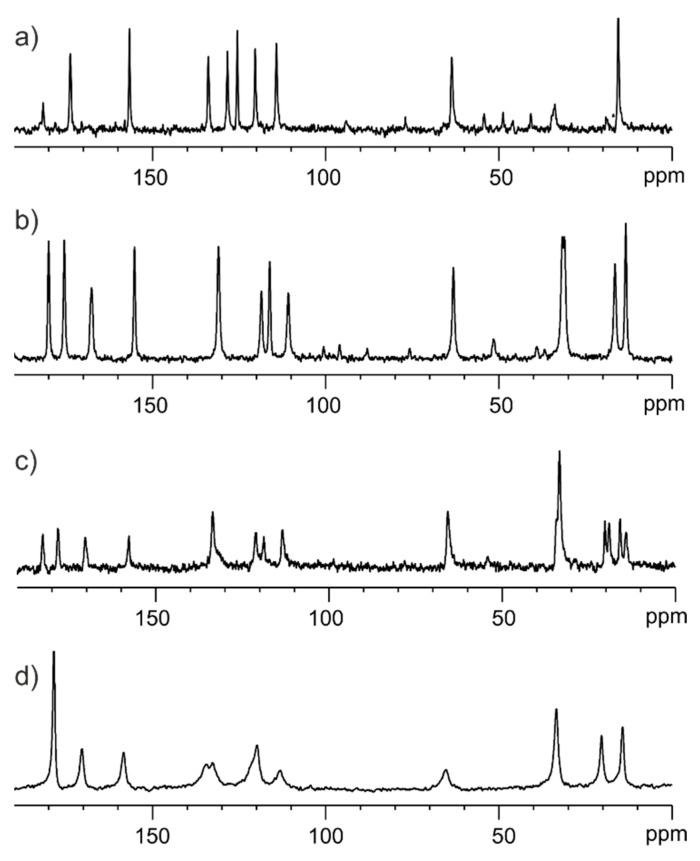
(**a**) ^13^C CP/MAS spectrum of physical mixture of ET and GLU, (**b**) ^13^C CP/MAS spectrum of well-defined ET:GLU cocrystal, (**c**) ^13^C CP/MAS spectrum of the physical mixture ET:GLU cocrystal and SiO_2(hydr)_ with a weight-to-weight ratio of 1:3, (**d**) ^13^C SPE/MAS spectrum of the ET:GLU/SiO_2(hydr)_ physical mixture (1:3) recorded with a spinning rate of 12 kHz.

**Figure 9 pharmaceutics-15-01234-f009:**
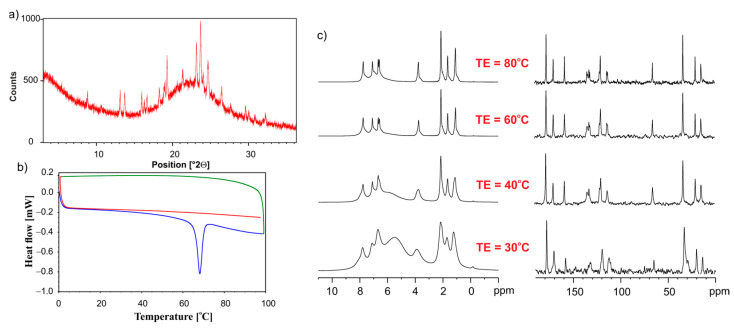
(**a**) PXRD recorded on a PANalytical 3 kW system in Bragg–Brentano geometry and with a Cu Kα (λ = 1.5425 Å) source, (**b**) the DSC measurement. The first heating run from 0 °C to 100 °C (blue line), then a cooling run from 100 °C to 0 °C (green line), followed by a second heating run from 0 °C to 100 °C (red line), (**c**) VT NMR data ^1^H MAS (left column) and ^13^C CP MAS (right column) recorded with a spinning rate of 12 kHz for the physical mixture of ET:GLU cocrystal and SiO_2(hydr)_.

**Figure 10 pharmaceutics-15-01234-f010:**
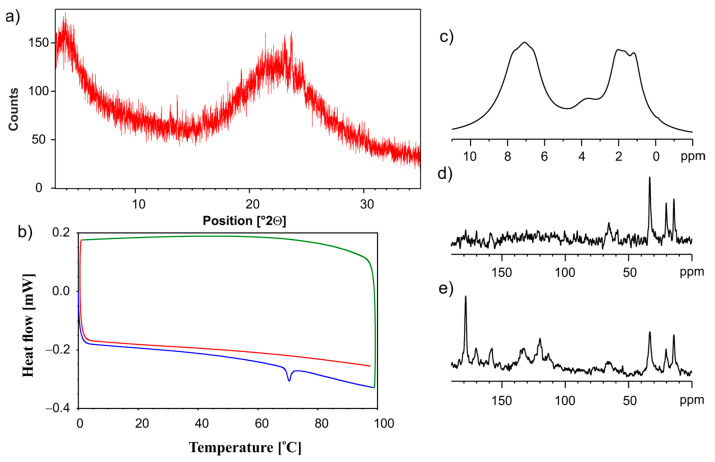
The (**a**) PXRD recorded on a PANalytical 3 kW system in Bragg–Brentano geometry and with a Cu Kα (λ = 1.5425 Å) source, (**b**) the DSC measurement. The first heating run from 0 °C to 100 °C (blue line), then a cooling run from 100 °C to 0 °C (green line), followed by a second heating run from 0 °C to 100 °C (red line), (**c**) ^1^H MAS, (**d**) ^13^C CP MAS and (**e**) ^13^C SPE MAS NMR measurements recorded with a spinning rate of 12 kHz for the physical mixture of ET:GLU cocrystal and SiO_2(hydr)_ after grinding in a ball mill.

**Figure 11 pharmaceutics-15-01234-f011:**
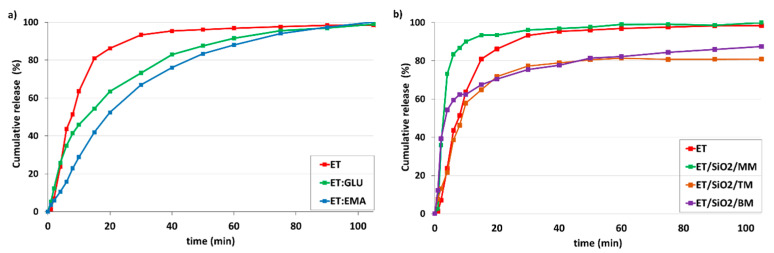
In vitro dissolution profiles of (**a**) ET and ET from ET:GLU, ET:EMA cocrystals, (**b**) ET and ET loaded in/on silica SiO_2_. Samples of ET in/on silica SiO_2_ were prepared by three methods as described in legend. Water with pH 5.7 was used as a dissolution medium. The estimated error in measurements was found to be +/−1%.

**Figure 12 pharmaceutics-15-01234-f012:**
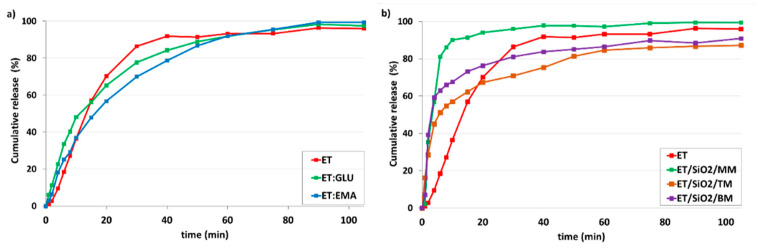
In vitro dissolution profiles of (**a**) ET and ET from ET:GLU, ET:EMA cocrystals, (**b**) ET and ET loaded in/on silica SiO_2_. Samples of ET in/on silica SiO_2_ were prepared by three methods as described in legend. Simulated gastric fluid without pepsin (SGFsp, pH 1.2) was used as a dissolution medium. The estimated error in measurements was found to be +/−1%.

**Figure 13 pharmaceutics-15-01234-f013:**
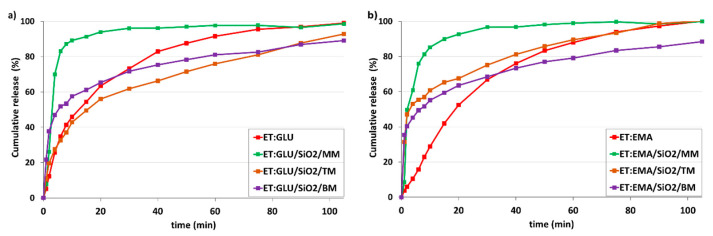
In vitro dissolution profiles of (**a**) ET in ET:GLU cocrystal and ET:GLU cocrystal loaded in/on silica SiO_2_, (**b**) ET in ET:EMA cocrystal and ET:EMA cocrystal loaded in/on silica SiO_2_. Samples of cocrystals loaded in/on silica SiO_2_ were prepared by three methods as described in legend. Water with pH 5.7 was used as a dissolution medium. The estimated error in measurements was found to be +/−1%.

**Figure 14 pharmaceutics-15-01234-f014:**
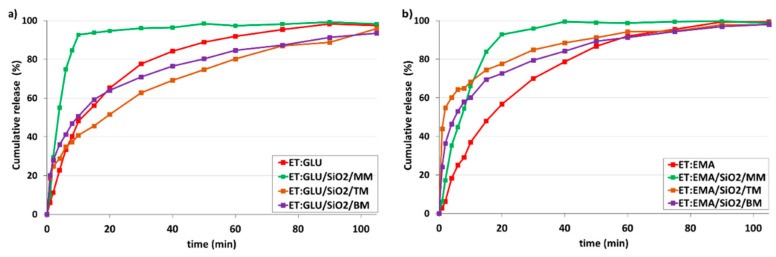
In vitro dissolution profiles of (**a**) ET in ET:GLU cocrystal and ET:GLU cocrystal loaded in/on silica SiO_2_, (**b**) ET in ET:EMA cocrystal and ET:EMA cocrystal loaded in/on silica SiO_2_. Samples of cocrystals loaded in/on silica SiO_2_ were prepared by three methods as described in legend. Simulated gastric fluid without pepsin (SGFsp) was used as a dissolution medium. The estimated error in measurements was found to be +/−1%.

## Data Availability

Not applicable.

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
