# Peer review of "Amorphization of Ethenzamide and Ethenzamide Cocrystals—A Case Study of Single and Binary Systems Forming Low-Melting Eutectic Phases Loaded on/in Silica Gel"

_pharmaceutics, 2023, doi:10.3390/pharmaceutics15041234_

Round 1

Reviewer 1 Report

The manuscript should be revised.

The authors must mention how they fixed the pH 5.7 for distilled water for dissolution tests because the water has not this pH. They should explain why they have chosen this pH. Similar tests must be done at least pH 6.8 (and 1.2).

The discussion of the DSC analysis must be revised (lines 252-258).

The authors wrote:

“Blue line represents the first run in temperature range from 0°C to 150°C with melting temperature (endothermic peak) at 132°C. The green line displays the profile for the cooling process.”

In the figure 5c there is not a blue line.

It is not correct to write “residual traces of crystalline ET”. The mass of ET in the mixture with silica is lower. It can be estimated the mass of crystalline ET from the mixture.

The discussion of the behavior of ET at higher temperature should be accompanied with the study of its biological activity.

The authors should explain why in the case of ET/SiO2/TM the dissolution curve reach only 80% of the drug dissolved in delivery medium.

The experimental points of the dissolution curves must have error bars.

Author Response

We are grateful to the reviewers for their comments and suggestions. We took this into account and modified the manuscript according to their requests.

Reviewer 1

The manuscript should be revised.

1) The authors must mention how they fixed the pH 5.7 for distilled water for dissolution tests because the water has not this pH. They should explain why they have chosen this pH. Similar tests must be done at least pH 6.8 (and 1.2).

Answer:

In our dissolution studies, we used water purified with the Milli-Q Direct Water Purification System. Such water has a pH of 5.7. This is a slightly lower value compared to distilled water. This information was corrected in the revised version of the manuscript.

As requested by the referee, in the revised version we have added a new set of dissolution results. We decided to take measurements at a pH of 1.2 (Simulated gastric fluid without pepsin (SGFsp)). The new kinetic profiles are shown in Figures 12 and 14. Our selection is based on the analysis of preliminary data and the conclusion that the profiles at pH = 5.7 and pH = 6.8 are very similar and do not provide important information about the tested systems. Comments describing the new results are given in the revised manuscript.

2) The discussion of the DSC analysis must be revised (lines 252-258).

The authors wrote: “Blue line represents the first run in temperature range from 0°C to 150°C with melting temperature (endothermic peak) at 132°C. The green line displays the profile for the cooling process.” In the figure 5c there is not a blue line.

Answer:

Unfortunately the blue and red lines are almost fully overlapped. We changed the figure and added dashed red line to better visibility.

3) It is not correct to write “residual traces of crystalline ET”. The mass of ET in the mixture with silica is lower. It can be estimated the mass of crystalline ET from the mixture.

Answer:

The phrase “residual traces of crystalline ET” in the revised version is replaced by the sentence "The sharp endothermic peak at 132°C represents the melting of trace amounts of crystalline ET, suggesting that amorphization is a continuous process and takes time".

4) The discussion of the behavior of ET at higher temperature should be accompanied with the study of its biological activity.

Answer:

The behaviour of ET and physiological pharmacokinetics at 37° C were exhaustively discussed in paper.

“Journal of Pharmacokinetics and Biopharmaceutics, Vol. 10, No. 6, 1982, 649-661.

Physiological Pharmacokinetics of Ethoxybenzamide Based on Biochemical Data Obtained in Vitro as well as on Physiological Data

Jiunn Huei Lin, Yuichi Sugiyama,  Shoji Awazu,  and Manabu Hanano”

This article has been added to the cited references.  Moreover, during the melting method of amorphization, we did not observe any traces of ET degradation, so the results published by Lin et al are also valid for our systems.

5) The authors should explain why in the case of ET/SiO2/TM the dissolution curve reach only 80% of the drug dissolved in delivery medium. The experimental points of the dissolution curves must have error bars.

Answer:

Explaining the problem with the ET/SiO2/TM dissolution profile and answering the question why 80% of ET is released for this system, we would like to emphasize that in the manuscript we presented data in the time range up to 100 minutes. 100% ET release is achieved after several hours of storage of samples in the delivery medium. It is worth noting that the difference in dissolution kinetics is visible when we compare the profiles in water (pH = 5.7) and gastric fluid (pH = 1.2).

Referring to the error in the measurements, we tested the reliability of our data. We found that this error is in the range of +/- 1%. A corresponding comment has been added to the legend of Figures 12-14.

Reviewer 2 Report

Dear Authors,

Please see the comments in the attached file.

Author Response

We are grateful to the reviewers for their comments and suggestions. We took this into account and modified the manuscript according to their requests.

Reviewer 2

In my opinion, the research procedure has been well planned, the techniques appropriately selected, and the article itself very well prepared. In addition, the results presented are valuable and provide new information in this area of knowledge. Therefore, I believe that with minor corrections, the paper can be published in the Pharmaceutics journal.

However, I encourage the Authors to address the following comments before publication:

  1. Section 3.1.2. – there are inaccuracies about the temperature ranges used; the Authors state that PXRD measurements were performed in the range of 20-100 °C, while in Fig. 6a diffractograms up to 120 °C are shown. On the other hand, Fig. 6b shows diffractograms for max. temp. of 140 °C, while the text mentions a range up to 120 °C. In my opinion, this should be sorted out.

Answer:

Thank you for pointing this out. We corrected the values in the text.

  1. Lines 339-341 – the Authors state: "Diffractograms shown in Figure 9a reveal that freshly mixed sample contains the crystalline ET:GLU components and amorphous phase but after melting and cooling sample is amorphous", meanwhile, the latter part of the sentence is not clear directly from the figure shown (there are no diffractograms for the sample after melting and cooling). I think this should be clarified in some way.

Answer:

The text has been changed.

  1. The same discussion of the results to Figure 9 is repeated twice in the paper (lines 338-346 and 353-361).

Answer:

You are right. We deleted the repeated sentences.

  1. Figures 11 and 12 – I think information on the repeatability of the results obtained / error bars would be highly recommended.

Answer:

Similar problem was addressed by referee 1. Please see our answer to this comment (point 5)

  1. In my opinion, it would be advisable to supplement the discussion of the results obtained by the Authors with a commentary on analogous studies on API amorphization conducted on other carriers, especially on silica other than silica gel (mesoporous silica, silica aerogels, etc.).

Answer:

A short discussion that links our previous results to current research has been added to Section Conclusions.

  1. „conformer” or „coformer” – the Authors should check the manuscript for this.

Answer:

Appropriate corrections have been made.